# Unlocking the mentally ill in Indonesia: An empirical study of the effectiveness of a *"Bebas Pasung"* program in Central Java

Tri Hayuning Tyas[1]*, Mary-Jo D. Good[2], Bambang Pratikno[3], M. A. Subandi[1], Carla R. Marchira[4], Byron J. Good[2]

**1** Faculty of Psychology, Gadjah Mada University, Yogyakarta, Indonesia, **2** Department of Global Health and Social Medicine, Harvard Medical School, Boston, Massachusetts, United States of America, **3** Soerojo Mental Hospital, Magelang, Indonesia, **4** Department of Psychiatry, Faculty of Medicine, Public Health, and Nursing, Gadjah Mada University, Yogyakarta, Indonesia

* t.h.tyas@ugm.ac.id

**Data Availability Statement:** For approved reasons, some access restrictions apply to the data underlying the findings. Given the sensitivity of the data and the fact that data are not formally de-

## Abstract

### Background

Locking or confinement of persons with severe mental illness has been common in Indonesia. In 2010, the Ministry of Health declared a policy that persons who were locked (*pasung*) should be unlocked or freed (*bebas*) from confinement and provided mental health services. This study is an empirical evaluation of the effectiveness of one *Bebas Pasung* program in Indonesia at two-year follow-up.

### Methods

From medical records in Soerojo Mental Hospital, Magelang, Central Java, 114 persons with severe mental illness who had been unlocked, treated, and returned to the community from four districts served by the hospital were identified. At two-year follow-up, 62 caregivers were able to be contacted and willing to participate in a study. Data were collected from hospital records about condition of the patient at time of "unlocking" and at discharge, and primary caregivers were interviewed about the previous locking and care of the patient since return, as well as experiences of caregiving.

### Results

We provide descriptive data concerning history of illness, reasons for locking, type of confinement, and care of the individual since return. 58% of those unlocked were men, 80% had diagnoses of schizophrenia, and mean age was 35. At follow-up, 24% of this sample had been re-locked; only 44% took medications regularly, including 33% of those re-locked and 47% of those not relocked. A majority cared for themselves, half were partially or fully productive, and the quality of life of family caregivers improved significantly since their family member was unlocked, treated, and returned home.

identified owing to on-going research, sharing of the data requires IRB approval for researcher who meet the criteria for access to confidential data. Data on the specific variables in this study are available upon request and requests may be sent to the Head of Research Ethics Committee, Faculty of Psychology, Universitas Gadjah Mada, email: komisietikpsi.psikologi@ugm.ac.id.

**Funding:** This research is supported by USAID Cooperative Agreement No. AID-497-A-11-00017, "Inter-University Partnerships for Strengthening Health Systems in Indonesia: Building New Capacity for Mental Health Care" (https://www.usaid.gov/id/indonesia). This was a USAID Cooperative Agreement with Harvard Medical School, with BJG as Project Director and MJDG as Co-Project Director. Subcontracts to Gadjah Mada University, with MAS and CRM as Principal Investigators, supported research activities in Yogyakarta from 2011 through 2014. Data analysis and writing was partially supported by a grant to the Department of Global Health and Social Medicine from the Harvard Medical School Center for Global Health Delivery–Dubai, for a project entitled "Building a Model for Comprehensive Mental Health Care in Yogyakarta, Indonesia." Harvard Principal Investigators were BJG and MJDG. Work in Indonesia was supported by a consortium agreement with Gadjah Mada University, with MAS and CRM as Principal Investigators. The funders had no role in study design, data collection and analysis, decision to publish, or preparation of the manuscript. There was no additional external funding received for this study.

**Competing interests:** The authors have declared that no competing interests exist.

## Conclusions

This program successfully unlocked, treated, and returned to their homes persons with severe mental illness living in *pasung* or restraints. Findings suggest such unlocking programs need to be linked more closely to community-based mental health and rehabilitation services, maintain care of the patient, and provide a path toward recovery.

## Introduction

The complex issues of institutional confinement of persons living with severe mental illness, as well as the use of seclusion and restraint within institutions, lie at the heart of the history of mental health services and efforts at reform. Although widely acknowledged and often vividly described, far less is known about the locking or confinement of persons with psychotic illness in community settings and families' homes. Over the last two decades, the rise of a renewed global mental health movement [1], of increased human rights attention to confinement of persons with severe mental illness in community settings [2] and of robust efforts within the WHO to develop specific guidelines for community based mental health care in low resources settings, launched initially as the Mental Health Gap Action Programme (mhGAP), have led to what might be called an "unlocking revolution" in some parts of the world. China and Indonesia serve as prominent examples [1–5].

In Indonesia, the term *pasung* is used to describe a wide range of forms of restraint, confinement and seclusion in community and home settings. These include use of wooden shackles, chains or tying with ropes, as well as seclusion in locked rooms in homes or separate spaces. Although use of *pasung* has been formally banned since 1977 in Indonesia, it is widely known to be highly prevalent, particularly among families with limited access to formal mental health services.

Genuine focus on directing Indonesian mental health services to "unlocking" is recent. Several turning points led both indirectly and directly to efforts to dramatically re-imagine mental health care in Indonesia, with renewed focus on trauma and explicit efforts to place unlocking and care for the severely mentally ill at the center of national mental health policy.

The violence associated with the fall of the Suharto regime in 1998, the ensuing ethnic and religious violence and the bombings in Bali [6, 7] and Jakarta [8] fostered a new interest in "trauma" and led to national and international efforts to extend mental health services beyond psychotic illness alone. The Great Indian Ocean Tsunami that struck the province of Aceh on December 26, 2004, killing over 160,000 persons, led to massive national and international responses, including psychosocial interventions. The tsunami brought to sharp attention a violent, often brutal conflict between the Indonesian military and the Free Aceh Movement underway in Aceh at the time of the tsunami. The Helsinki Agreement [9] ended that conflict, and mental health activities were extended to those affected by the conflict as well. While trauma was one focus of the response to this dual disaster in Aceh, projects led by the Ministry of Health, as well as those supported by the International Organization for Migration, used the opportunity to develop far more comprehensive models of mental health services than had been present to date in any part of Indonesia, leading to the evolution of what became described as the "Aceh model" of community-based services [10–12]. Unlocking of persons with severe mental illness and bringing them into care was one explicit policy of the provincial government of Aceh [13].

In 2010, the Division of Mental Health of the Ministry of Health of Indonesia, led by Dr. Irmansyah, launched the 2010 *Bebas Pasung* ('Free from *Pasung*') initiative, aimed at drawing attention to the plight of thousands of persons with severe mental illness in Indonesia living in *pasung*, bringing into view what was considered a major human rights issue and focusing new attention on freeing persons living in restraints in the community and bringing them into care [5, 14, 15]. This policy initiative was largely aspirational; it was turned into legally binding regulations. However, this initiative was a critical turning point in imagining the future of mental health services for Indonesia, making unlocking of persons in *pasung* an explicit focus of mental health policy and research. It was further supported by the passage of Indonesia's first real national mental health law in 2014, passage of a National Disability Law in 2016, Indonesia's ratification of the internationally recognized Convention on the Rights of Persons with Disabilities, and the broadening of the "Standard Minimal Services" to be provided by all districts through their primary care system to include care of persons with severe mental illness.

Given the decentralized nature of health policy and services implementation in Indonesia, the 2010 *Bebas Pasung* initiative was implemented with varying levels of intensity and in quite different ways across the very diverse provinces and districts of the country. A recent systematic review of the research on *pasung* in both English and Indonesian makes it clear both that persons living in *pasung* remain widespread in Indonesia and that systematic data about its prevalence are quite limited [5]. In 2014, the Ministry of Health of Indonesia estimated lifetime prevalence of persons experiencing *pasung* as 57,000 persons and point prevalence as 18,800. Hidayat et al. [5] suggest that given how families conceal persons living in *pasung*, this was likely a serious underestimation. They also argue that the government's claim that the Free Pasung Program reduced point prevalence to 12,200 by 2018 is open to serious doubt. The Human Rights Watch [2] report on shackling in Indonesia makes it clear that confinement in *pasung* remains a persistent problem in Indonesia, as well as just how little we know about the nature and prevalence of the phenomenon. In 2017, the Indonesian Ministry of Health release regulations that were meant to reduce the use of *pasung* for those with mental illness and support unlocking programs through coordination and cooperation between the central and provincial governments. However, the regulations did not provide for legal consequences.

The Free Pasung Program unleashed not only a series of local initiatives, associated with provincial and district governments and national and regional mental hospitals, but also led to a significant body of new research. Hidayat et al. [5] identify 34 published research articles on *pasung* in Indonesia in English and Indonesian through 2019. Of these, 21 are qualitative reports, and studies are judged to be of variable quality. The Ministry of Health claimed that 10% of persons in *pasung* were hospitalized and treated between 2009 and 2014. However, the review finds that "there is no data on how many of those PWMD [Persons With Mental Disorders] were successfully supported through rehabilitation or returned to pasung in the community." [5]. The overall finding of the review (p. 16) states this even more forcefully. "In Indonesia, the Bebas Pasung Program. . . involves the provision of community-based mental health services alongside intensive education campaigns. None of the articles measured the effectiveness of the program or how the program is delivered."

This article provides just such data about one program developed as part of the larger Bebas Pasung Initiative. It briefly describes the implementation of the *bebas pasung* policy in Central Java's primary psychiatric hospital, *Rumah Sakit Jiwa Soerojo* (Soerojo Mental Hospital), and provides findings from a two-year follow-up study of persons unlocked, treated in the hospital, and returned to their families. The study asks how effective the unlocking program was, what its strengths and weaknesses were, and how family caregivers experienced the program and evaluated its efficacy.

## Methods

The study reported here was a two-year follow-up study, using mixed methods, of one specific *Bebas Pasung*, or unlocking, program, carried out by the Soerojo Mental Hospital, in Magelang, Central Java, beginning in 2012. The study gathered data concerning 62 individuals who had been locked, treated in Soerojo Hospital, and released back into families and communities. The study was designed to provide important descriptive data about the history of illness of those locked, how the individual had come to be locked, the form of *pasung*, and the patient's condition at time of hospitalization and after return to the family. It also focused explicitly on who constituted the caregivers, what their experience of caregiving had been prior to unlocking, and how gender of patient and caregiver influenced that experience. The study was designed to evaluate the effectiveness of the intervention program, including how many persons had been relocked since returning home, factors related to such relocking, and the caregivers' overall experience of the program. The findings are reported using descriptive statistics, such as percentages, means, and standard deviations. A repeated-measures ANOVA in SPSS Statistic version 25 is used to analyzed the time sequence data, followed by post-hoc pairwise comparison with an LSD adjustment to compare the differences of conditions at the three points of time. A one-way ANOVA is also used to analyze the differences of conditions between the relocked and no relocked status at each time point.

The study was approved by the ethics review boards of Harvard Medical School and Soerojo Mental Hospital's research ethics committee.

### The Central Java intervention project: Procedures of the unlocking and treatment program

In 2012, the Governor of the Province of Central Java announced a policy of *bebas pasung*, requiring that intersectoral programs at provincial and district levels should be developed to implement the policy. He declared that relevant ministries, health service providers, and hospitals should be responsible for implementing this policy and that the Provincial Health Office should be responsible for reporting the services they are providing to the office of the Governor. The policy mandated four types of activities: case-finding; provision of basic and referral health care; provision of rehabilitation services, using community-based as well as government services; and facilitation of the return of individuals who had been locked to their families and communities. Specific regulations for implementing these services were not developed. Importantly, the policy did not stipulate long-term or continuing services to be provided to individuals and families.

The Director of the central provincial mental hospital, *Rumah Sakit Jiwa Soerojo* (Soerojo Mental Hospital) in Magelang, made the decision that his hospital should take the lead in developing a program to implement the Governor's mandate and that services should be provided to unlocked individuals free of charge. Earlier in 2011, the hospital director made the hospital's Department of Public Mental Health (*Bagian Kesehatan Jiwa Masyarakat*, shortened routinely to *Bagian Keswamas)* responsible for developing and carrying out the policy, in collaboration with the clinical departments. This department, directed by a highly committed nurse, developed procedures and implemented the policy in several steps. First, the Department began a series of outreach activities into local communities. Outreach teams were developed to go into communities, meet with district and village officials and leaders to provide information about the program, provide community education through direct educational programs and by working with local women's organizations, and produce radio messages to educate the community about the nature of severe mental illness, the importance of unlocking, and the promise of free health services for those who were unlocked and brought to Soerojo

Hospital for care. Although radio messages were broadcast throughout the region and services were to be provided free of charge to persons from throughout Central Java, a province with 35 districts with a population of over 32 million people, direct outreach services focused primarily on those districts closest to the city of Magelang.

Second, "evacuation teams," consisting of nurses, doctors, other hospital staff, and ambulance drivers from Soerojo Hospital, were developed to participate in releasing individuals who were locked and helping transport them to the hospital. Referrals requesting services were made to the hospital staff by family members, community leaders, primary health care center staff, staff from the Social Welfare ministry, and lay persons through a "hotline" provided by the hospital. Members of the staff of the hospital would verify the referral and arrange for an "evacuation" of the individual. Medical staff participated, along with local community members, because many of those brought to the hospital were in poor physical and mental health condition.

Third, individuals were admitted into the hospital and provided with services in the medical, psychiatric, and rehabilitation units. The patients' conditions and services were recorded as part of the routine medical records of the hospital. Patients were often hospitalized in Soerojo Hospital—or other state mental hospitals—for one to three months to stabilize their medical condition and drug levels. Medical and psychosocial rehabilitation services were provided, including gardening in the hospital grounds, handicraft activities, and, for a limited number of patients, metal work and bicycle or motorcycle repair.

Finally, return to the community and to families was organized by hospital staff. Family members were contacted, invited to come to the hospital, meet with the hospital staff briefly, be given medications (usually for one month) and instructed on usage, and take the patient home. If families were unable to pick up patients, in a limited number of cases patients were returned using hospital ambulances or other resources. Families and patients were invited to return to the hospital's outpatient services for routine care and medications, or alternatively, to go to their local primary health care centers for medications. The program was scaled up rapidly, and in March of 2013 Soerojo Hospital reported that 1,135 individuals had been released from *pasung*, treated in Soerojo or one of the three other provincial mental hospitals, and returned to their communities.

**The follow-up study design.** The study gathered data concerning the status of the patient and caregivers' experiences at time of unlocking (T1), the time of return of the patient to the family or community post-treatment (T2), and the time of the follow-up study which lasted from June 8, 2013 to September 24, 2013 (T3). Information from hospital medical records and description of the condition of the patient and family at the time of unlocking and at time of release from the hospital by members of the evacuation team and persons participating in the discharge were recorded by the research team. Stories and specific information were also gathered through a long interview with the individual's primary caregiver, asking retrospectively about the history of illness and locking, characteristics of the patient at both the time of unlocking and the time of return to the family, utilization of health services and medication, family experience throughout this process, and the status of the patient and medical care utilization since discharge from the hospital and specifically at the time of the follow-up interview.

**Sample.** The sample was developed by preparing a complete listing of individuals who participated in the unlocking program (N = 114) from four districts—the city and the district of Magelang, in which the hospital is located, and two adjacent districts, requiring travel of up to approximately three hours—along with the contact information of the "guardian" responsible for the hospitalization of the individual. An effort was made to contact all 114 guardians, to describe the study, and ask permission to visit the family. If guardians were family members,

permission was requested directly to visit the family. If guardians were medical or social welfare staff or community leaders, they were asked to contact the family and ask for permission to contact them directly or visit. In some cases, the guardians could not be contacted. In some cases, the families declined to make a time to visit to participate in the study or the patient was no longer living with the family. When families were visited, a primary caregiver was identified to participate in the study. Inclusion criteria included that the caregiver be at least 19 years old, be cognitively and physically able to participate in the study, and be a primary caregiver at least since the time of the unlocking. The purpose of the study was described, the caregiver was asked to give signed consent, and the individual with mental illness, when available, was asked to give oral consent. In the end, 68 families were visited, and complete data were collected for 62 individuals who were unlocked and their families. Visits, including explaining the study, being given informed consent, participating in completing the family questionnaire and the formal interview, and socializing lasted from two to four hours. First author and or one of the nurse asked the participants to fill out the family questionnaire or read the questionnaire out loud and asked participants to respond. Interviews were conducted by the first author, by one of four nurses trained to carry out the interviews, or some grouping of more than one of these. In addition, the first author and/or one of the nurse interviewers returned to a subsample of the families' homes for further, in-depth observation and discussion, leading to 24 intensive case studies.

**Instruments.** The research instrument was developed in part based on the instrument used for a related follow-up study of persons unlocked during China's "686 Program" [3] also conducted jointly with the Harvard Medical School authors of this study. Two instruments were developed: one to record information from hospital records and medical staff participating in the evacuation and discharge of the patient, and one to guide the interview with the family caregiver and gather both qualitative and quantitative data. The first instrument was used to record hospital record information about the condition, diagnosis, and treatment of the patient at the time of admission, and comparable data about the patient at time of discharge. In addition, information about the condition of the patient and the setting of the locking at the time of the evacuation was recorded by a member of the evacuation team, and data about the quality of the patient and family at the time of the discharge was recorded by a person involved in the discharge. A second form was developed to guide the interview and record information from the caregiver at the time of the follow-up study. Narrative data were elicited to record specific information about the history of illness and treatment, reasons and nature of locking, and family burden and experience, as well as information on symptoms and social functioning of the patient. A format for recording data on locking was developed, using the previous research experience of the first author on *pasung* [16]. Scalar information about caregivers' experience of caring for the ill family member was based on an instrument developed for gathering similar data about family experience in a study of unlocking conducted in China [3]. Family members were asked to rate their subjective experiences on analogue scales from 0 "no impact" to 5 "extremely negative impact" for five categories of family burden: stigma or shame (*malu*), psychological pressures, economic burden, loss of personal or family spirit or energy (*semangat*), and interpersonal relationships. Ratings concerning T1 and T2 were obtained by caregivers' retrospective reflections on period before unlocking, experience at time of return of the family member, as well as current subjective experience (T3). Questions about social functioning were developed based on the authors' previous research in both Aceh and Yogyakarta. Interviews with caregivers were recorded and transcribed. In-depth interviews for the intensive case studies were also recorded and transcribed.

## Results

In what follows we provide some of the most salient findings about those who had been unlocked through the program, about who the caregivers were and what their experiences were, and about the effectiveness of the program both in objective terms and subjective terms from the point of view of caregivers. Because most data on *pasung* in Indonesia are case reports or qualitative data [5], here we provide primarily quantitative data, rather than analysis of the in-depth interviews with family members.

### Characteristics of the individuals, caregivers, and forms of confinement at time of unlocking

The research team recorded gender and age of the persons who were locked, as well as of their primary caregivers, data on who were primary givers for whom, and self-described family social class. Thirty one (50%) of the families who participated in the research identified themselves as economically "poor" or "very poor", while the remaining half identified themselves as "average". Of the patients who had been in pasung (N = 62), 58% were men, 42% women. The patients had a mean age of 35 (SD = 8.9) and ranged from 20 to 69 years of age (28%were aged 20–29, 45% aged 30–39, and 29% aged 40–69). Of the 56 persons for whom a hospital diagnosis was available, 45 (80%) were diagnosed with schizophrenia and 8 (14%) with schizoaffective disorder. 26% of the 62 of the patients in the study sample were reported to have been ill for ten or more years; 47% (56% of men, 35% of women) had been ill two years or less. About 78% of the men and 27% of the women had some history of having previously worked.

Caregivers were 55% men, 45% women. The mean age of the caregivers was 53 years old (SD = 13.7). Their ages ranged from 20 to 80, with 82% of caregivers over 40 years of age.). 58% of all caregivers were parents, 26% siblings or cousins, 5% were children of the individual in *pasung*, 5% were aunts or uncles, 3% spouses, and 3% were persons from the community.

In this sample, women were only slightly more likely to be primary caregivers for women patients (45%), men slightly more likely to be primary caregivers for male patients (61%).

There were a range of forms of pasung used to restrain these patients at the time they were unlocked. The largest number (61%) were locked into a room—which in some cases was simply the common room in the house, in other cases a tiny, dirty room with only a bed without blankets and a tiny toilet. The remainder were chained (36%), tied by rope, or detained in classic *pasung* or wooden stocks. The great majority (86%) were restrained within the family's house; 16% were locked in a space in a separate building, and a small number were restrained some place nearby. Just over 70% were locked constantly, 24 hours a day/7 days per week, while the remaining were locked only part of the time.

At the time those who were in *pasung* were released and taken to the hospital, the "evacuation team" asked family members a series of questions about why the individual had been locked. 75% of men and 42% of women were said to be locked because they had become violent; a slightly larger number were reported to have threatened others (83% of men, 42% of women); and even more (83% of men and 50% of women) were reported to have "run *amuk*," a phrase used to describe persons who become totally out of control. A significant number of men (56%) and nearly all women (88%) had run away or threatened to run away; and 31% of men and 46% of women were considered a threat to harm themselves.

A significant number of family members reported, as an important reason for locking, that there was no one to care for the individuals (for 44% of men and 73% of women); family members reported feeling hopeless as a reason for locking (of men 56%, women 73%); and a large number gave economic reasons for locking the individuals (for 50% of the men, 73% of the women). A relatively small number reported not agreeing with the medications given (17% for

men, 23% for women), and a significant number (25% for men, 27% for women) reported they felt the individual would be helped by being locked.

These overall findings for the reasons families give for locking a member suffering from severe mental illness are quite consistent with the primary types of reasons given in the qualitative studies and case reports described in the literature review [5] and are also consistent with the stories family caregivers gave in published narrative accounts [17–23] and the in-depth case studies to members of the research team for this project, as well as stories the UGM-Harvard research team members have heard over the years.

Families take no pleasure in having had to lock one of their members; they are often filled with shame and hide the family member from others in the community and members of the health care system. But they tell of feeling driven to the use of *pasung* by the individual's violence to family and community members, especially for men, and fear of harm to the individual, particular young women, if they run away and wander in the community or go even further afield. And they report not having enough family members to care for the individual and having to detain the individual while family members work to earn their living.

The hospital evacuation teams recorded the condition of the person living in *pasung* at the time of their unlocking. 44% were in extremely bad condition, dirty, and living in a totally unsanitary room; an additional 45% were described as being in slightly less dirty and unsanitary spaces, or "not so clean". Only 11% were reported to be in reasonably good sanitary and health conditions. This is one reason hospitalization often required extensive medical care and physical rehabilitation as well as the provision of mental health services. When asked the reasons for requesting that a family member be unlocked and taken to the hospital, 50% hoped for effective care, 42% for free medication, and 23% hoped for continuous care. Only 4 of the 62 (6.5%) described being forced to participate in the program.

## Findings concerning effectiveness of the program

This project studied the effectiveness of the *bebas pasung* program in Soerojo Hospital in two formal ways: by examining how many of those who were released and treated, before returning home, were re-locked at some time during the two years since unlocking; and by providing scalar data from the primary about a number of dimensions of their experience before unlocking, at the time of the individual's return to their home, and at the two year follow-up interview.

Overall, as Table 1 shows, 24% of those unlocked—17% of men, 24% of women—were reported to have been returned to *pasung* at some time during the past two years since being returned to the family. This may also be framed by noting that 76% of those unlocked remained unlocked through the initial two years of being returned to the community. Of those relocked, use of chains or mechanical restraints was less, while more were confined to a room in the house, than was true before the intervention.

The scalar data (Tables 2 to 6) show significant improvement in the experience of the family care providers from the time of unlocking (T1) to the time of return home (T2), and demonstrates that this improvement was largely maintained or even extended through the following

**Table 1. Numbers of patients relocked during two year period by gender.**

| Pasung status/ patient's gender | Male N(%) | Female N(%) | Total |
|---|---|---|---|
| Relocked at some time following unlocking | 6 (17%) | 9 (35%) | 15 (24%) |
| Not relocked since being returned | 30 (83%) | 17 (65%) | 47 (76%) |
| Total N | 36 (100%) | 26 (100%) | 62 (100%) |

**Table 2.  Family's level of shame (*malu*) at different points of time (N = 62).**

| Points of time | Mean (SD) | Points of time | Mean difference | p values |
|---|---|---|---|---|
| T1 | 3.29 (1.50) | T1-T2 | 1.50 | .000 |
| T2 | 1.79 (1.04) | T2-T3 | 0.29 | .008 |
| T3 | 1.50 (.099) | T1-T3 | 1.79 | .000 |

T1 = while in pasung; T2 = after discharged from hospital; T3 = 18–24 months later
(5-extremely negative impact; 1 = no impact)

**Table 3.  Family's economic burden at different points of time (N = 62).**

| Points of time | Mean (SD) | Points of time | Mean difference | p values |
|---|---|---|---|---|
| T1 | 2.89 (1.66) | T1-T2 | 1.05 | .000 |
| T2 | 1.84 (1.12) | T2-T3 | 0.02 | .898 |
| T3 | 1.82 (1.11) | T1-T3 | 1.07 | .000 |

T1 = while in pasung; T2 = after discharged from hospital; T3 = 18–24 months later
(5-extremely negative impact; 1 = no impact)

**Table 4.  Family's psychological distress at different points of time.**

| Points of time | Mean (SD) | Points of time | Mean difference | p values |
|---|---|---|---|---|
| T1 | 3.66 (1.38) | T1-T2 | 1.76 | .000 |
| T2 | 1.90 (1.16) | T2-T3 | 0.24 | .087 |
| T3 | 1.66 (1.04) | T1-T3 | 2.00 | .000 |

T1 = while in pasung; T2 = after discharged from hospital; T3 = 18–24 months later
(5-extremely negative impact; 1 = no impact)

**Table 5.  Patient's relationship to family at different points of time (N = 62).**

| Points of time | Mean (SD) | Points of time | Mean difference | p values |
|---|---|---|---|---|
| T1 | 2.16 (1.48) | T1-T2 | 0.64 | .000 |
| T2 | 1.52 (1.02) | T2-T3 | 0.13 | .020 |
| T3 | 1.39 (.95) | T1-T3 | 0.77 | .000 |

T1 = while in pasung; T2 = after discharged from hospital; T3 = 18–24 months later
(5-extremely negative impact; 1 = no impact)

**Table 6.  Family's spirit (*semangat*) at different points of time (N = 62).**

| Points of time | Mean (SD) | Points of time | Mean difference | p values |
|---|---|---|---|---|
| T1 | 2.79 (1.65) | T1-T2 | 1.08 | .000 |
| T2 | 1.71 (.98) | T2-T3 | 0.26 | .003 |
| T3 | 1.45 (.84) | T1-T3 | 1.34 | .000 |

T1 = while in pasung; T2 = after discharged from hospital; T3 = 18–24 months later
(5-extremely negative impact; 1 = no impact)

**Table 7. Family's experience of shame (*malu*) with or without re-pasung/relocked.**

| Points of time | Mean scores | Mean scores | p values |
|---|---|---|---|
| | No relocking | Relocking | |
| T1 | 3.23 | 3.47 | F = .271 (p = .600) |
| T2 | 1.53 | 2.60 | F = 14.60 (p < .000) |
| T3 | 1.17 | 2.53 | F = 33.04 (p < .000) |

T1 = while in pasung; T2 = after discharged from hospital; T3 = 18–24 months later

(5-extremely negative impact; 1 = no impact)

two years (T3). This was true as reported by both male and female caregivers. Whether in terms of their experience of shame, psychological distress reported, economic burden reported, relationships within the family, or overall family energy or "spirit" (*semangat*), caregivers reported great benefits from the unlocking program.

This improvement was, not surprisingly, more powerful for those caregivers who did not feel it necessary to re-lock the person who had been unlocked and treated (Table 7) than for those who found it necessary to relock their family member. But even for these caregivers, there remained a benefit along the scalar dimensions studied for the program.

At the time the patient was returned to the family, 80% were reported to take medications regularly. However, at follow-up, 44% of all patients continued to take medications regularly and another 21% irregularly or frequently not taking medications. Table 8 shows the relationship between continued use of medications and whether the individual was returned to *pasung* of confinement during the two-year follow-up program. Of those relocked, 33% took medications regularly and 67% either irregularly or stopped medications completely, while of those not relocked during the two-year period, 47% took medications regularly and 53% either irregularly or stopped medications entirely. There was thus a tendency for those who did not take medicines to be at higher risk for being relocked, but the difference is not statistically significant.

At follow-up, 51% of this group of persons who had been confined in *pasung* two years earlier were now able to care for themselves, 24% were reported to be capable of productive work and another 25% of more limited work, 32% were reported to have good relationships with their family and another 50% fair relations with their family, and 25% were reported to have good relationships with neighbors while 38% were reported to be fairly able to engage other others in the community.

## Discussion

This study provides new empirical evidence concerning persons with severe mental illness who have been placed in *pasung* or family-based confinement in one province in Indonesia,

**Table 8. Medication status of patients relocked (Re-pasung).**

| Medication Status | Relocked N(%) | Not relocked N(%) | Total |
|---|---|---|---|
| Regularly taking medication up to present | 5 (33%) | 22 (47%) | 27 (44%) |
| Irregularly taking medication | 2 (13%) | 11(23%) | 13 (21%) |
| Frequently not taking medication | 3 (20%) | 4 (9%) | 7 (11%) |
| Began, but stopped taking medication | 3 (20%) | 5 (11%) | 8 (13%) |
| Never take medicine | 2 (13%) | 5 (11%) | 7 (11%) |
| Total N | 15 (100%) | 47 (100%) | |
| | (24%) | (76%) | 62 (100%) |

the nature of the locking or confinement, reasons families gave for locking, who were considered the primary caregivers, and families' ratings of their experiences associated with having a member of their family in *pasung*. The fact that 70% of patients were considered "dirty and unsanitary" at the time of the unlocking provides a numerical rating of the conditions of persons who lived in confinement or *pasung*. Photos taken by the hospital teams responsible for the unlocking and for taking those unlocked to the hospital tell this story far more vividly. Many lived in genuinely horrifying conditions and required very significant medical care alongside the mental health care they received when they were unlocked and hospitalized. The *bebas pasung* program thus had a genuine sense of "freeing" (*bebas*) those confined in often inhumane living conditions.

There were clear strengths and weaknesses of the Soerojo Mental Hospital program of unlocking and mental health treatment described in this study. Overall, the program was developed systematically, freed 1,135 individuals with severe mental illness from locking by March 2013, and provided them with good quality, hospital-based medical care and initial rehabilitation services, free of cost, before returning them to their families. The program provided a powerful learning experience by the public health team of the Soerojo Hospital, and the program continues until today although with a different program title.

The program also had a major effect on the health of the patients and the experience of family members. Three-quarters of all persons unlocked had not been re-locked at any time during the two years since unlocking, and family caregivers experienced enormous benefits of the program. Over half of those persons who had been locked, shackled or confined were now able to care for themselves, and nearly half were able to engage in fully or partially productive work. Over 80% were now reported to have good or fair relations with their family and over 60% able to fully or partially engage with neighbors. Given these findings, it is not surprising that caregivers rated their experiences of living with and caring for the ill member significantly improved at the time the person was returned from the hospital, and that positive rating of their experiences continued into year two when the follow-up was conducted.

On the other hand, only 44% of these persons took medications regularly at two-year follow-up, and another 21% reported taking medications irregularly. Of these, 47% of those who had never been relocked, compared with 33% of those who were relocked, were taking medications regularly at time of follow-up, and 70% of those who had never been relocked, compared with 47% of those who were relocked, were taking medications regularly or irregularly at time of follow-up. These findings testify to the limited access to continuous care and medications. They do not provide strong evidence that failure to take medications is the primary reason for being re-locked. The findings show that following unlocking, treatment, and return to the community, the use of metal chains or other mechanical restraints was less for those who were relocked, and more were confined to a separate room in the house. This may indicate a different purpose for restraining the individual or change of the family's perceptions towards the person in pasung, which led them to choose a somewhat "better" form of restraint.

Medication compliance may be linked directly to the leading complaint about the program by caregivers—that there was too little follow-up by the program. The quality of continued community-based mental health services varied by locale and activity of local primary health care centers, and when services were provided by visiting nurses these were commented on positively. On the other hand, only a third of caregivers rated services of village health workers (*cadre)* positively. This indicates that the continuation of community-based mental health services by the mental health workers is critically important, especially to support the caregivers in providing care, to ensure treatment adherence, and to promote recovery for the person with mental illness. The active involvement of community in the mental health program may help

the family to ease the burden from stigma and help the patient to be in supportive environment.

The widely cited follow-up study of an unlocking intervention in China [3], which served as one important model for this follow-up study, found that only 8% of those unlocked had been relocked by year 7, and that 72% remained on medication. Compared with these data, the unlocking program in Central Java was less successful. It is important to note that the Chinese program was part of a comprehensive mental health reform movement that led to regular visits by mental health teams down to the village and family level. It is equally important to note that the Chinese program—the so-called 686 Program [24, 25]—was initiated with the rationale of reducing the "dangerousness" of persons with severe mental illness, so that levels of stigmatizing surveillance were quite high and remain high until today.

By contrast, programs that resulted from the 2010 national call for an Indonesia free of *pasung* operated, to a large extent, as stand-alone programs, not as a comprehensive reform of the overall mental health system. The Indonesian program described here was not linked to the development of comprehensive, community-based outreach programs to provide services for those living in the community with severe mental illnesses. This limited the overall effectiveness of the program. In addition, although the district offices of the Ministry of Social Welfare are legally responsible for providing all rehabilitation services, rather than those of the Ministry of Health, Social Welfare programs are largely focused on physical rehabilitation. Despite these limitations, the program made an enormous difference in the lives of many persons in this part of Central Java who had been living in *pasung*, as well as for their families.

Limitations of this study include the fact that it was only able to gather data from 62 of the 114 target families, leading to a potential selection biases of the study. Our ability to generalize findings to the entire population released from pasung and returned to the community by this program is thus somewhat limited. The follow-up period of this study was only 18 to 24 months, leaving additional important questions about longer term outcomes. And the study was unable to conduct a direct evaluation of the level of symptoms of the patients at the time of the home visit, limiting what we can say about actual outcomes. Instead, we focused on issues of relocking as the primary outcome of this study. More comprehensive and longer-term follow-up studies of this program and unlocking programs around Indonesia are needed.

## Conclusions

The unlocking and treatment programs initiated by Ministry of Health of Indonesia in 2010 have varied in structure and quality. As we have noted, following the only major review of unlocking in Indonesia [5], we need far more empirical data about local programs and their short- and long-term effectiveness. This study was carried out to provide basic empirical information about persons with severe mental illness in one area of Indonesia who lived in community-based confinement, about one specific program of unlocking and treatment, and about the effectiveness of this program over a two-year period. This study provides evidence of the enormous importance of such programs. Despite its limitations, the program had enormous benefits for those living with severe mental illness who were in *pasung* and for their families. This study also documents the limitations of programs not embedded in more comprehensive mental health services reform. Clearly, unlocking programs in Indonesia need to be linked far more closely to the development of real community-based mental health and rehabilitation services that outreach care and support for of patients and families and provide a genuine pathway toward recovery [26].

## Acknowledgments

The authors gratefully acknowledge the full support of many staff members from the Soerojo Mental Hospital, Magelang, Central Java, Indonesia. Special thanks to the Keswamas staff members: Jofita Panggelo, Noviandy Radhikabudi, and the four nurses from Soerojo Hospital who participated in the interview process. We are particularly grateful to the patients and families who actively participated in this project.

We acknowledge that the four scales on family experience were based on the research in China, described by Guan et al. (2015). Prof. Mary-Jo D. Good was a primary advisor on the project in China, where she participated in designing the research instrument. Parts of this instrument were adapted for use in the Central Java project, for which Prof. Good was also primary advisor.

## Author Contributions

**Conceptualization:** Tri Hayuning Tyas, Mary-Jo D. Good, M. A. Subandi, Carla R. Marchira, Byron J. Good.

**Data curation:** Tri Hayuning Tyas, Bambang Pratikno.

**Formal analysis:** Tri Hayuning Tyas, Mary-Jo D. Good, M. A. Subandi, Byron J. Good.

**Funding acquisition:** Mary-Jo D. Good, M. A. Subandi, Carla R. Marchira, Byron J. Good.

**Investigation:** Tri Hayuning Tyas, Bambang Pratikno.

**Methodology:** Tri Hayuning Tyas, Mary-Jo D. Good, M. A. Subandi, Byron J. Good.

**Project administration:** Bambang Pratikno.

**Resources:** Bambang Pratikno, M. A. Subandi, Carla R. Marchira, Byron J. Good.

**Supervision:** Mary-Jo D. Good, M. A. Subandi, Carla R. Marchira, Byron J. Good.

**Validation:** Tri Hayuning Tyas.

**Writing – original draft:** Tri Hayuning Tyas, Mary-Jo D. Good, Byron J. Good.

**Writing – review & editing:** Tri Hayuning Tyas, Mary-Jo D. Good, M. A. Subandi, Carla R. Marchira, Byron J. Good.

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
