## [Decision Letter · Decision Letter 0]

23 Apr 2024

PONE-D-24-12316Unlocking the mentally ill in Indonesia: An empirical study of the effectiveness of a “Bebas Pasung” program in Central JavaPLOS ONE

Dear Dr. Tyas,

Thank you for submitting your manuscript to PLOS ONE. After careful consideration, we feel that it has merit but does not fully meet PLOS ONE’s publication criteria as it currently stands. Therefore, we invite you to submit a revised version of the manuscript that addresses the points raised during the review process.

We look forward to receiving your revised manuscript.

Kind regards,

Ashish Wasudeo Khobragade, MD

Academic Editor

PLOS ONE

Journal Requirements:

‘”This research is supported by USAID Cooperative Agreement No. AID-497-A-11-00017, "Inter-University Partnerships for Strengthening Health Systems in Indonesia: Building New Capacity for Mental Health Care" (https://www.usaid.gov/id/indonesia ). This was a USAID Cooperative Agreement with Harvard Medical School, with BJG as Project Director and MJDG as Co-Project Director. Subcontracts to Gadjah Mada University, with MA Subandi (MAS) and Carla R. Marchira (CRM) as Principal Investigators, supported research activities in Yogyakarta from 2011 through 2014. Data analysis and writing was partially supported by a grant to the Department of Global Health and Social Medicine from the Harvard Medical School Center for Global Health Delivery--Dubai, for a project entitled "Building a Model for Comprehensive Mental Health Care in Yogyakarta, Indonesia." Harvard Principal Investigators were BJG and MJDG. Work in Indonesia was supported by a consortium agreement with Gadjah Mada University, with MAS and CRM as Principal Investigators. The funders had no role in study design, data collection and analysis, decision to publish, or preparation of the manuscript.”

3. In the online submission form you indicate that your data is not available for proprietary reasons and have provided a contact point for accessing this data. Please note that your current contact point is a co-author on this manuscript. According to our Data Policy, the contact point must not be an author on the manuscript and must be an institutional contact, ideally not an individual. Please revise your data statement to a non-author institutional point of contact, such as a data access or ethics committee, and send this to us via return email. Please also include contact information for the third party organization, and please include the full citation of where the data can be found.

Additional Editor Comments (if provided):

It is my pleasure to evaluate the manuscript titled “Unlocking the mentally ill in Indonesia: An empirical study of the effectiveness of a “Bebas Pasung” program in Central Java”. In this manuscript, the effectiveness of the 'Bebas Pasung' program is studied by the authors. The manuscript is interesting and of public health significance, but it requires revision.

1. The total number of tables in the manuscript is 16. Findings from some tables (those with fewer rows and columns) can be mentioned in the result section so that the number of tables can be reduced. Cite all the tables within the text.

2. The statistical tests used are not mentioned in the manuscript. Information on statistical analysis is missing in the methodology section.

Reviewers' comments:

Reviewer's Responses to Questions

**Comments to the Author**

1. Is the manuscript technically sound, and do the data support the conclusions?

Reviewer #1: Yes

Reviewer #2: Yes

2. Has the statistical analysis been performed appropriately and rigorously? 

Reviewer #1: Yes

Reviewer #2: Yes

3. Have the authors made all data underlying the findings in their manuscript fully available?

Reviewer #1: Yes

Reviewer #2: No

4. Is the manuscript presented in an intelligible fashion and written in standard English?

Reviewer #1: Yes

Reviewer #2: Yes

5. Review Comments to the Author

Reviewer #1: 1. Is the manuscript technically sound, and does the data support the conclusions?

The manuscript appears to be technically sound. It describes a rigorous empirical study evaluating the effectiveness of the Bebas Pasung program in Central Java, Indonesia, for individuals with severe mental illness. The methods employed for data collection, including interviews with caregivers and analysis of hospital records, seem appropriate and thorough for the study's goals. The results, which provide descriptive data on patients' histories, reasons for confinement, types of confinement, and caregivers' experiences, support the study's conclusions. The study successfully demonstrates that the program has been effective in unlocking, treating, and reintegrating individuals with severe mental illness back into their communities. Furthermore, the manuscript outlines the program's strengths and weaknesses, showing a balanced view supporting the conclusions drawn.

2. Has the statistical analysis been performed appropriately and rigorously?

The manuscript includes detailed statistical analysis, particularly in the results section, where scalar data from caregivers is presented. Using mean differences and p-values to assess changes in caregivers' experiences of shame, economic burden, psychological distress, family relationships, and overall spirit indicates a rigorous statistical approach. However, the manuscript could benefit from more detailed information on the statistical methods used for analysis, such as specific tests performed. The statistical analysis seems appropriately performed and supports the study's findings.

3. Have the authors made all data underlying the findings in their manuscript fully available?

The manuscript details the data collection process from hospital records and caregiver interviews, providing a solid foundation for the findings presented. However, it does not explicitly state if all the raw data underlying the findings, such as interview transcripts or detailed hospital record data, are fully available for external review. For transparency and to strengthen the study's credibility, the authors should clarify the availability of the underlying data.

4. Is the manuscript presented in an intelligible fashion and written in standard English?

The manuscript is well-structured, with transparent sections for the abstract, introduction, methods, results, discussion, and conclusions. Using tables to present data enhances readability and the understanding of critical findings. The manuscript is written in standard English, with a formal tone appropriate for a scientific audience. Minor editorial revisions could further refine the manuscript's clarity and presentation quality.

Summary and Recommendations

The manuscript provides a valuable contribution to the field of mental health, particularly in the context of community-based care for individuals with severe mental illness in low-resource settings. To strengthen the manuscript further, the authors could consider:

• Providing more detailed descriptions of the statistical methods used.

• Clarifying the availability of underlying data for review by others.

• Undergoing minor editorial revisions for clarity and presentation quality.

Overall, the manuscript meets the criteria for a technically sound piece of scientific research, with data that support the conclusions drawn.

Reviewer #2: Thank you for the opportunity to review this manuscript. I am pleased to provide my feedback on this insightful and valuable piece of research.

The topic of the manuscript is highly significant and addresses an under-discussed issue within international mental health care, specifically the practice of pasung in Indonesia. The authors have highlighted a crucial social and health issue that remains largely hidden from the international community. The study not only sheds light on the challenges and needs of persons with severe mental illness who have been subjected to confinement but also emphasizes the transformative impact of Indonesia's Bebas Pasung program.

I commend the authors for their diligent efforts in collecting and presenting data from a difficult-to-reach population. Their findings provide a meaningful contribution to our understanding of the longitudinal outcomes for those affected by severe mental illness after their release from confinement. The descriptive data offered about the history of illness, reasons for locking, and types of confinement enriches the discourse on mental health services in culturally diverse settings.

Additionally, I am personally grateful for the opportunity this review process has afforded me to learn and understand more deeply about the phenomenon of pasung. Reading this manuscript has been truly enlightening. I believe its publication will significantly benefit the broader psychiatric community by illuminating a previously obscured aspect of mental health care in a unique setting. Parallels to similarly tragic issues are faced also in Western contexts, now and in the past, underscoring the global relevance of this research.

I fully support the publication of this manuscript. It is an important contribution that not only informs but also prompts important discussions and potential policy changes in the treatment of mental health globally.

I have a few suggestions that could further enrich the discussion and implications of your findings:

- Could you provide a brief explanation of the Bebas Pasung policy, focusing on its legal, administrative, and organizational aspects, in the introduction?

- Could you elaborate on how your data might be interpreted within the broader context of potential future interventions aimed at enhancing compliance and preventing re-locking?

- Based on your data, what types of support or interventions do you suggest could be implemented to improve both the family's circumstances and patient outcomes?

- Is it possible to compare your findings with data from a "typical" psychiatric population within your country to highlight any unique trends or differences?

- Are there more socio-economic data available about the families involved in the study that could be shared?

- Do you have additional clinical data about the studied population that you could share (e.g. CGI-S scores at program entry and CGI-I scores at exit)?

- Additionally, would it be possible to anonymize the data using a feasible method to facilitate sharing with fellow researchers?

Also, to enhance the methodological rigor of your results, I suggest incorporating a brief statistical methods paragraph that outlines the statistical significance tests you've conducted.

Thank you once again for the opportunity to review this significant work. I look forward to its publication and the continued dialogue it will undoubtedly inspire within the mental health community.

Best regards

6. PLOS authors have the option to publish the peer review history of their article (what does this mean?). If published, this will include your full peer review and any attached files.

Reviewer #1: No

Reviewer #2: **Yes: **Pierfelice Cutrufelli

---

## [Author Response · Author response to Decision Letter 0]

16 Jul 2024

Journal requirement:

1. We have attempted follow all style requirements, including file naming.

2. We have revised the Funding Statement and include it in the cover letter as suggested. Here is the full version of Funding Statement: 

This research is supported by USAID Cooperative Agreement No. AID-497-A-11-00017, "Inter-University Partnerships for Strengthening Health Systems in Indonesia: Building New Capacity for Mental Health Care" (https://www.usaid.gov/id/indonesia ). This was a USAID Cooperative Agreement with Harvard Medical School, with BJG as Project Director and MJDG as Co-Project Director. Subcontracts to Gadjah Mada University, with MAS and CRM as Principal Investigators, supported research activities in Yogyakarta from 2011 through 2014. Data analysis and writing was partially supported by a grant to the Department of Global Health and Social Medicine from the Harvard Medical School Center for Global Health Delivery--Dubai, for a project entitled "Building a Model for Comprehensive Mental Health Care in Yogyakarta, Indonesia." Harvard Principal Investigators were BJG and MJDG. Work in Indonesia was supported by a consortium agreement with Gadjah Mada University, with MAS and CRM as Principal Investigators. The funders had no role in study design, data collection and analysis, decision to publish, or preparation of the manuscript. There was no additional external funding received for this study. 

3. We have made arranged for the Research Ethics Committee of the Faculty of Psychology, Gadjah Mada University, to be responsible for making the data available according to current norms. 

Here is the full Data Availability Statement: 

For approved reasons, some access restrictions apply to the data underlying the findings. Given the sensitivity of the data and the fact that data are not formally de-identified owing to on-going research, sharing of the data requires IRB approval for researcher who meet the criteria for access to confidential data. Data on the specific variables in this study are available upon request and requests may be sent to the Head of Research Ethics Committee, Faculty of Psychology Universitas Gadjah Mada, email: komisietikpsi.psikologi@ugm.ac.id

4. Interview transcripts and detailed hospital record data cannot be made available due to sensitive information that may jeopardize the identity of participants. However, data on the specific variables in this study are available upon request. 

5. We have reviewed the reference list for completeness and format. None of the cited papers have been retracted, to our knowledge.

Additional Editor Comments

We appreciate the overall positive tenor of the comments, and we have responded to the two specific suggestions:

1. We have placed all of the data describing the sample from the first eight tables into the text. All remaining eight tables are cited in the text.

2. We have added information about the statistical analyses in the Methods Section (in lines 157-162).

Review comments by Reviewer 1 and 2 

Basic comments (1-4) by both reviewers.

We very much appreciate the overall positive tenor of this review.

1. Is the manuscript provided technically sound and does the data support the conclusions?

Response: no additional revisions are suggested. 

2. Has the statistical analysis been performed appropriately and rigorously? 

Comments: However, the manuscript could benefit from more detailed information on the statistical methods used for analysis, such as specific test performed. 

Response: As described above, added information concerning statistical methods used for analysis has been added in lines 157-162.

3. Have the authors made all data underlying the findings in their manuscript fully available? Response: As described in response to the Journal Requirements item 3, we have now made arrangements for the de-identified data to be made available in a fashion that meets the PLOS Data policy that the Research Ethics Committee of the Faculty of Psychology of Gadjah Mada University will be the responsible sight for the data. The hospital records cannot be made available, given confidentiality of clinical data, and the interview transcripts cannot be adequately de-identified. However, all data coded and drawn from these sources will be available on request to the Research Ethics Committee office.

4. Is the manuscript presented in an intelligible fashion and written in standard English?

Comments: Minor editorial revisions could further refine the manuscript’s clarify and presentation quality.

Response: The manuscript had undergone additional editorial revision for clarity and presentation quality by the primary American PIs on the original study, both of whom have been journal editors.

Summary and Recommendations

Reviewer #1

Again, we deeply appreciate the fact that this reviewer finds the results and analyses to be of importance for understanding the care of the mentally ill in low resource settings and for the development of further research aimed at improving policies and services.

Suggestions:

Provide more detailed descriptions of the statistical methods.

We have responded to this above; added information provided in lines 157-162.

Clarify data availability

 See above.

Undergo minor editorial revisions

 See above.

Reviewer #2

We deeply appreciate this reviewer's overall comments about the importance of this study and what this reviewer learned personally from this manuscript. We agree, of course, that although very different, Western societies face equally tragic lack of genuine care for persons with severe mental illness. In the United States, we recognize that "pasung" or confinement often means the carceral system or homeless shelters, and that "return to the community" often has extraordinarily tragic outcomes. 

1. Could you provide a brief explanation of the Bebas Pasung policy, focusing on its legal, administrative and organization aspect in the introduction?

Response: We have added information concerning the Bebas Pasung policy both at the national level and in the Province of Central Java in lines 98-99 and 119-123. Indonesia’s Bebas Pasung initiative was launched in 2010 by the division of Mental Health of the Ministry of Health of Indonesia. This initiative was not followed by a formal policy or regulations that were binding. There is no obligation for provincial or district governments to respond. As noted, the 2010 aspirational policy declaration was followed by the passage of Indonesia's National Mental Health Law in 2014. However, that law was never implemented or turned into regulations, as required by the law, and has now been replaced by what is known as the Omnibus Bill. Fortunately, the provincial government of Central Java responded by the development of a provincial initiative, that led to the Bebas Pasung Program described in this study.

2. Could you elaborate on how the data might be interpreted within the broader context of potential future interventions aimed at enhancing compliance and preventing re-locking?

Response: Full response to this issue would and perhaps will require a full paper, further intervention trials and research, and many years of continued work. We have discussed the relevance of this study for overall mental health services policy and design, adding further detail, in line 475-480, 486-491, and 501-510 of the Discussion and lines 530-533 of the Conclusion.

Potential future interventions aimed at enhancing treatment compliance and reducing relocking as much as possible include strengthening the community-based mental health services that promote community-based rehabilitation programs, with a focus on further training of Community Mental Health Nurses in the primary care system and added training and funding of village mental health cadre. 

3. Based on your data, what types of support or interventions do you suggest could be implemented to improve both the family’s circumstances and patients’ outcomes? 

Response: Again, as noted in response to comment 2, we have attempted to address these issues in broad terms. A longer response would include the importance and limitations of the National Health Insurance system (e.g. the fact that its implementation allows mental hospital to limit hospital stays to as little as 7-10 days), the need for recovery-oriented training of physicians and nurses in the national primary health system, and also financial support for families caring for members with severe mental illness. Obviously, many of these issues are beyond the scope of this research report. We appreciate this reviewer's concern for these issues. However, we have tried to focus on the specific findings of this study and the immediate implications for mental health policy.

4. Is it possible to compare your findings with data from a “typical” psychiatric population within your country to high light any unique trends or differences? 

Response: It is somewhat unclear what this suggestion means. We believe the psychiatric population of Central Java is not atypical, although Indonesia is vastly varied in local cultural differences and in access to services. Far more studies are needed to give a more comprehensive view of conditions around Indonesia. We have gestured to this issue both by noting the recent literature review of pasung in Indonesia and by noting the importance of far more research in this domain.

We also feel that the explicit comparison of the data from this study with data from a much longer study in China, in which the overall PIs of this program of research were also involved, adds significantly to our ability to speak to both the strengths and weaknesses of this program in Indonesia.

5. Are there more socio-economic data available about the families involved in the study that could be shared?

Response: We responded to this by adding information in lines 304-307. We asked the family caregivers how they defined their economic condition. 50% said they were poor or very poor, the remaining half said they were average. 

6. Do you have additional clinical data about bout the studied population that could shared? (e.g. CGI-S scores at program entry and CGI-I scores at exit?)

Response: Unfortunately, we were not able to make any further evaluation of current psychiatric symptom levels of patients at the time of the home interviews. This is of course an additional limitation of the study. We added a note about this in lines 514-518.

7. Additionally, would it be possible to anonymize the data using a feasible method to facilitate sharing with fellow researchers? 

Response: Yes. We have made arrangements to have data made available, as described above. 

8. To enhance the methodological rigor of your results, I suggest incorporating a brief statistical methods paragraph that outlines the statistical significance test you’ve conducted. 

Response: We have done so as noted above.

---

## [Decision Letter · Decision Letter 1]

5 Aug 2024

Unlocking the mentally ill in Indonesia: An empirical study of the effectiveness of a “Bebas Pasung” program in Central Java

PONE-D-24-12316R1

Dear Dr. Tyas,

We’re pleased to inform you that your manuscript has been judged scientifically suitable for publication and will be formally accepted for publication once it meets all outstanding technical requirements.

Kind regards,

Ashish Wasudeo Khobragade, MD

Academic Editor

PLOS ONE

Additional Editor Comments (optional):

Reviewers' comments:

Reviewer's Responses to Questions

**Comments to the Author**

1. If the authors have adequately addressed your comments raised in a previous round of review and you feel that this manuscript is now acceptable for publication, you may indicate that here to bypass the “Comments to the Author” section, enter your conflict of interest statement in the “Confidential to Editor” section, and submit your "Accept" recommendation.

Reviewer #1: All comments have been addressed

Reviewer #2: All comments have been addressed

2. Is the manuscript technically sound, and do the data support the conclusions?

Reviewer #1: Yes

Reviewer #2: Yes

3. Has the statistical analysis been performed appropriately and rigorously? 

Reviewer #1: Yes

Reviewer #2: Yes

4. Have the authors made all data underlying the findings in their manuscript fully available?

Reviewer #1: Yes

Reviewer #2: Yes

5. Is the manuscript presented in an intelligible fashion and written in standard English?

Reviewer #1: Yes

Reviewer #2: Yes

6. Review Comments to the Author

Reviewer #1: Introduction and Background: The background information on the historical and cultural context of pasung in Indonesia is comprehensive. The discussion on the policy developments and the Bebas Pasung initiative provides a clear understanding of the study's importance.

Methods: The methodology is robust, with clear descriptions of the sample selection, data collection, and analysis processes. The use of both qualitative and quantitative methods enriches the study's findings.

Results: The results section is detailed, providing valuable data on the demographics of the participants, the types of confinement, and the outcomes of the program. The use of tables to present scalar data on the caregivers' experiences before and after the program is effective.

Discussion: The discussion effectively interprets the findings, highlighting both the strengths and weaknesses of the program. The comparison with similar programs in other countries, such as China, adds depth to the analysis.

Limitations: The authors acknowledge the study's limitations, including the potential for selection bias and the inability to conduct direct evaluations of current psychiatric symptoms at the time of follow-up. This transparency is commendable.

Recommendations: The recommendation for integrating unlocking programs with comprehensive community-based mental health services is well-founded. Future studies could explore long-term outcomes and more extensive follow-up to build on the findings presented here.

Data availability: I am aware the authors have attempted to make the underlying data available ("Interview transcripts and detailed hospital record data cannot be made available due to sensitive information that may jeopardize the identity of participants. However, data on the specific variables in this study are available upon request." I will leave this up to the editor to justify if this is appropriate enough for publication.

On all other accounts, this is a valuable and well written manuscript.

Reviewer #2: Dear Authors,

I am personally satisfied with the status of the revised work. Thank you for the insights regarding the psychiatric situation in analogous contexts in the West. It is important to highlight that stigma and marginalization are culturally influenced, but not culturally defined; unfortunately, no culture is exempt from perpetuating them.

Regards.

7. PLOS authors have the option to publish the peer review history of their article (what does this mean?). If published, this will include your full peer review and any attached files.

Reviewer #1: **Yes: **Harry James Gaffney

Reviewer #2: **Yes: **Pierfelice Cutrufelli

---

## [Editor Report · Acceptance letter]

22 Aug 2024

PONE-D-24-12316R1 

PLOS ONE

Dear Dr. Tyas, 

I'm pleased to inform you that your manuscript has been deemed suitable for publication in PLOS ONE. Congratulations! Your manuscript is now being handed over to our production team.

Kind regards, 

on behalf of

Dr. Ashish Wasudeo Khobragade 

Academic Editor

PLOS ONE